# Comparative Phytochemical Analyses of Flowers from *Primula veris* subsp. *veris* Growing Wild and from Ex Situ Cultivation in Greece

**DOI:** 10.3390/foods12132623

**Published:** 2023-07-06

**Authors:** Konstantia Graikou, Anna Mpishinioti, Nikolaos Tsafantakis, Eleni Maloupa, Katerina Grigoriadou, Ioanna Chinou

**Affiliations:** 1Laboratory of Pharmacognosy and Chemistry of Natural Products, Faculty of Pharmacy, National and Kapodistrian University of Athens, Panepistimiopolis, Zografou, 15771 Athens, Greece; kgraikou@pharm.uoa.gr (K.G.); annampisxinioti@gmail.com (A.M.); ntsafantakis@uoa.gr (N.T.); 2Balkan Botanic Garden of Kroussia-Laboratory for the Conservation and Evaluation of Native and Floricultural Species, Institute of Plant Breeding and Genetic Resources, Hellenic Agricultural Organization—DIMITRA, Thermi, P.O. Box 60458, 57001 Thessaloniki, Greece; emaloupa@elgo.gr (E.M.); katgrigoriadou@elgo.gr (K.G.)

**Keywords:** *Primula veris* subsp. *veris*, infusion, ex situ cultivation, flavonoids, UHPLC-HRMS, TPC, antioxidant activity

## Abstract

In the last decades, *Primula veris* subsp. *veris* (roots and flowers) has been over harvested through legal and illegal ways in Greece, due to its extremely high commercial demand, as it is used in industry because of its well-known therapeutic properties. As ex situ cultures of the plant have been already developed, in the current comparative study, the herbal teas (infusions) from both flowers of cowslip growing wild in the Prespa Lake Park (NW Greece), and from ex situ propagated and cultivated plant material, have been investigated, with the ultimate goal of assessing them qualitatively. Furthermore, through classic phytochemical studies, the ten most abundant metabolites, belonging to the chemical categories of flavonol-glycosides and methoxy flavones, have been identified and structurally determined. The chemical profile of both infusions has been further analyzed through UHPLC-HRMS, showing that they show only light differences. The total phenolic content (TPC) of both studied samples (wild and ex situ cultivation), was determined by the Folin–Ciocalteau method, followed by an antioxidant activity assay though DPPH where, in both cases, wild plants exerted higher phenolic load and stronger antioxidative properties. According to the reported results, it could be proposed that the ex situ cultivated plant material could facilitate the mass production of plants and the sustainable cultivation of cowslip in the Greek mountains.

## 1. Introduction

Several edible flowers have been consumed traditionally since antiquity, among which are roses, violets and lavender, mostly to add flavor and color to foods. Moreover, it has been suggested that their chemical composition and potential health promoting effects could be further explored [1]. The primrose botanical family (Primulaceae) comprises of approx. 400 species widespread from Anatolia, Caucasus, Europe and the Himalayas to Western China. *Primula veris* L. subsp. *veris* is an evergreen herbaceous perennial plant native to temperate regions of Europe and Asia, with deep yellow flowers which are produced in spring and have a characteristic pleasant odor similar to vanilla [1]. The herb is commonly known as cowslip or cowslip primrose, a member of the Greek flora (mostly the mountainous and semi-mountainous regions of the mainland), and has extensive traditional use. Both roots and flowers of the herb in the form of herbal teas (infusions), as expectorant for cough and cold, have been used widely since antiquity, in the same manner towards respiratory disorders, as well as in other preparations considered as dietary supplements [1,2]. In the Middle Ages, it was known as St. Peter’s herb or Petrella, and was sought after by Florentine apothecaries, and has been recommended by Hildegard von Bingen (in the book Physica CCIX) for its medicinal use only externally, while its leaves were also consumed as food and as a flavoring agent in wines. The etymology of *Primula* derives from the Latin for “first”, which refers to the cowslip’s early flowering. There are references to cowslips in the works of William Shakespeare (*Henry V*, *The Tempest* and *A Midsummer Night’s Dream*), while its flowers have been used in several ceremonies to symbolize winning grace, youth and healing [3,4,5,6].

Further data on the medicinal use of cowslip’s flowers in Europe exist since the beginning of the 20th century [7], while the medicinal potential of *P. veris* is adequately supported by the European Medicines Agency’s Committee on Herbal Medicinal Products, in which the flower’s preparations have specific indications as expectorant for the relief cough associated with cold. The mode of the expectorant action of *Primula* saponins is not satisfactorily clarified through in vivo studies on pharmacological/toxicological effects of extracts from cowslip flower which showed a significant increase in the production of bronchial secretion, while in the United States, *P. veris* flowers are classified as Class 1 botanicals, which means they can be safely consumed when used appropriately [7].

There are numerous plant species (about 30,000) considered edible (used as food, food supplements, flavorings or colorants) worldwide, but only a small portion of species have been studied until now, the majority still remaining an unexplored source. Edible plants and, especially, their flowers are of increasing popularity in food industry, as they are a rich source of nutrients and phytochemicals, such as phenolics, vitamins and minerals, and known for their high nutritional value and health benefits [8].

The use of edible flowers is well-accepted and approved historically in different cultures, with the most well-known examples being the flowers of lavender (*Lavandula angustifolia*), rose (several *Rosa* species), jasmine (*Jasminus officinalis*), violet and mallow (*Malva sylvestris*), which have been widely used in salads, soups, drinks and desserts, mostly as flavoring [1].

Regarding the edible value of the *Primula* species, it is adequately appreciated that the beverages (infusion) from the leaves of *P. veris* and *P. elatior* [9], as well as the flowers from *P. veris* Demasi et al., 2021 [8], contain high quantities of vitamin C, which is very useful in the human diet. Moreover, the enrichment of the diet with edible flowers as sources of antioxidants has been studied and proven to exert beneficial effects on human health, as *Primula* species have been shown to consist of high amounts of phenolic compounds [10].

The cowslip’s flowers have been used for skin-related disorders [11] and, furthermore, as enrichment in saponins and in polyphenolic compounds in both white and red wines, which are effectively extracted into the wine, and constitute a highly pro-health component of diets, valuable in promoting heart’s health and preventing against cardiac failure [12].

According to the literature, the chemistry of the plant has been investigated, proving its flowers as rich source of phenolic type of metabolites in the forms of both aglycons and glycosides (such as apigenin, quercetine, kaemferol, isorhamnetin, cynaroside, rutin, hyperoside, etc.), as well as polymethoxylated flavonoids acting as chemotaxonomic markers for the genus *Primula* (8-methoxy-flavone; 3′,4′methylenedioxy-5′-methoxyflavone, etc.). Several triterpenic saponins are the dominating constituents in the roots of the species, contributing to the expectorant activity of the herb [13,14,15,16,17,18,19,20,21,22].

Very recently, herbal extracts of cowslip have exerted a cardioprotective effect when administered in animals with experimental chronic heart failure, by increasing the myocardial contractility [23,24], as well as anti-influenza virus [25] potency and anti-inflammatory properties [26,27].

In Greece, cowslips are often harvested directly from the wild due to their well-known therapeutic properties, and the plant material is sold in national and international markets. Due to this practice, there has been an increased commercial demand for the crude plant material, which has triggered the uncontrolled over-harvesting of wild plant populations in Greece and in other neighboring countries, such as Bulgaria, mostly in illegal ways [28,29].

In the last five years, large quantities (hundreds of kilograms) of illicit cowslip collections (*P. veris* subsp. *veris*) have been seized by the Greek authorities, especially in the region of north-west Macedonia, intended to be exported illegally to other Balkan countries (mostly Albania) and, from there, to be trafficked and, as a result, the herb’s population is dangerously decreasing. As cowslip is a valuable plant species with declining populations, efforts were needed to preserve the natural environment as well as crops for its sustainable use. Towards this direction, a successful attempt was the ex-situ production of *Primula veris* subsp. *veris* at the Hellenic Agricultural Organization-Demeter in 2020 [25], following an optimized protocol for the micropropagation of cowslip via shoot tip explants derived from seedlings, in order to facilitate the mass production of plants and the sustainable cultivation of the plant species. This approach has also been followed by the cultivation of roots [30,31].

Set in the northwestern corner of Greece, where three countries Greece, Albania and North Macedonia come together around two lakes, the Prespa National Park (PNP) covers an area of approx. 257 km^2^. It extends attitudinally from 850 m at the levels of the lakes (Great and Lesser Prespa) to 2.330 m on mount Varnous. The lands surrounding the basin are characterized by considerable geological and biological interest and, especially, the Prespa Lakes, the highest tectonic lakes in the Balkan peninsula, are characterized by rare local flora, including protected species [2].

In 2009, 11 smaller areas of the PNP were designated “Protected Natural Formation or Landscape Elements”, while the whole PNP has an unusually high number of plant species; up to November 2011, more than 1800 species and subspecies of vascular plants have been recorded, per unit area. The local flora consists of species unique to the Greek flora, others under international protection, and/or local species potentially threatened with extinction, like cowslips [32,33].

In the framework of our studies on selected herbs and bee-keeping products from the geographical part of northwest Greece (Prespa National Park, PNP), which is characterized as a plant endemism center and biodiversity hotspot [28], we present herein the first comprehensive report of chemical profile of the traditionally consumed herbal tea (infusion), according the EMA’s mode of preparation, from the Greek native *Primula veris* subsp. *veris* (PVPinf), growing wild in the highly protected PNP (NW Greece), in comparison with the same preparation from ex situ cultivation (PVDinf) in collaboration with Hellenic Agricultural Organization -Demeter, Institute of Plant Breeding and Genetic Resources. UHPLC-HRMS, as well as classical chromatographic methods of isolations and further spectral analyses, were used.

## 2. Materials and Methods

### 2.1. Chemicals and Reagents

All the chemicals and reagents were purchased from: Merck (Darmstadt, Germany) (ethanol absolute, Folin–Ciocalteu reagent, gallic acid, sodium carbonate (Na_2_CO_3_)), Carlo Erba Reagents (Val-de-Reuil, France) (dimethyl sulfoxide (DMSO)), Fisher Scientific (Loughborough, UK) (methanol high-performance liquid chromatography (HPLC)-grade), and Glentham Life Sciences (Corsham, UK) (2,2-diphenyl-1-picrylhydrazyl (DPPH•), stationary phase microcrystalline cellulose (cellulose microcrystalline, Merck), and SephadexLH-20 polyhydroxypropylated dextran gel stationary phase 25–100 µm (Pharmacia Fine Chemicals, Darmstadt, Germany).

### 2.2. Plant Material

The crude plant material was collected from the Prespa Lake Park during the flowering period in 2021, and was botanically identified by F.N. Sakellarakis, Section of Ecology and Systematics, National and Kapodistrian University of Athens, Greece, while the flowers from the ex-situ propagated and cultivated plant material were provided by Dr E. Maloupa and Dr K. Grigoriadou from the Hellenic Agricultural Organization-Demeter (2022). After ex situ seed germination trials and preliminary cultivation of the seedlings until maturation, a voucher specimen was deposited in the herbarium of the BBGK, Institute of Plant Breeding and Genetic Resources (HAO-DEMETER), under the code 105,428.

The flowers of both plant samples were kept in a shadowy and dry environment and were grounded by a laboratory mill and stored in darkness at room temperature.

### 2.3. Preparation of Infusions (Water Extract)

The preparation of infusions followed the EMA’s monograph on traditional herbal medicine (150 mL boiled water were added in 1 gr of dried grounded flowers) [6], followed by filtration and lyophilization.

The infusion of growing wild *P. veris* flowers (9.85 gr) afforded after lyophilization 4.83 gr, while the infusion of ex situ cultivated *P. veris* subsp. *veris* flowers, 2.4 gr afforded after lyophilization 1.18 gr. Then, the samples were studied phytochemically through open column chromatography and NMR spectroscopy.

### 2.4. UHPLC-HRMS Analysis

Analyses were performed on an UHPLC-HRMS/MS Orbitrap Q-Exactive platform (Thermo Scientific, San Jose, CA, USA). A full scan with a mass range of 100–1200 Da on a centroid mode was applied, while HRMS data (70,000 resolution) were collected in both negative and positive ionization modes under the following conditions: capillary temperature, 320 °C; spray voltage, 2.7 kV; S-lense Rf level, 50 V; sheath gas flow, 40 arb. units; aux gas flow, 5 arb. units; and aux. gas heater temperature, 50 °C.

A Hypersil Gold UPLC C18 (2.1 × 100 mm, 1.9 μm) reversed phase column (Thermo Fisher Scientific, San Jose, CA, USA) was used for the separations. The mobile phase consisted of solvents A: ultra-pure H_2_O 0.1% (*v*/*v*) FA, and B: ACN. A 16 min gradient method was used, varying as follows: 0–1 min, 5% B (isocratic gradient); 1–11 min, 5–95% B (linear gradient); 11–14 min, 95% B (isocratic gradient, column cleaning); 14–14.1 min, 95–5% B (linear gradient); and 14.1–16 min, 5% B (isocratic, column equilibration). The flow rate was 0.260 mL/min, and the injection volume was 5 μL. The column temperature was kept at 40 °C, while the sample tray temperature was set at 10 °C.

### 2.5. Fractionation, and Purification Procedures

PVP (600 mg) has been subjected in column chromatography with microcrystalline cellulose and eluted with CH_3_COOH 15%; 160 fractions (A1-A160) have been collected.

Fraction A10-A19 (73 mg) has been subjected to preparative TLC cellulose plate (CH_3_COOH 30%) to afford clitorin (kaempferol-3-O-(2, 6-α-L-dirhamnopyranosyl-β-D glucopyranoside) (**7**) (12.1 mg). Fraction A20-A40 (219 mg) has been subjected to preparative TLC cellulose plate (CH_3_COOH 30%) to afford quercetin-3-O-β-neohesperidoside (**6**) (2 mg) and kaempferol-3-O-β-neohesperidoside (**5**) (1.5 mg).

Furthermore, the fraction A45-96 (148 mg) has been subjected to column chromatography Sephadex LH-20 eluted with methanol to afford 10 fractions (B1-B10). Fraction B2-B4 (4.5 mg) was identified as isorhamnetin-3-O-β-glucopyranosyl-(1-2) -β-glucopyranosyl-(1-6)-β-glucopyranoside (**4**). Fraction B5-B9 (35 mg) has been subjected to preparative TLC cellulose plate (CH_3_COOH 30%) and the one band was a mixture (2.3 mg) of metabolites quercetin-3-O-β-glucopyranoside (**1**) and isorhamnetin-3- O-β-gluco pyranoside (**2**), respectively, and the other band another mixture (5.7 mg) of quercetin-3-O-β-glucopyranosyl-(1-2)-β-glucopyranosyl-(1-6)-β-glucopyranoside (**3**) and isorhamnetin-3-O-β-glucopyranosyl-(1-2)-β-glucopyranosyl-(1-6)-β–glucopyranoside (**4**).

Fraction A100-A142 (18 mg) has been subjected to column chromatography with microcrystalline cellulose eluted with cyclohexane: EtOAc (gradient 100:0 up to 100% EtOAc) to give 20 fractions (C1-C20). Fractions C4-C10 (9.8 mg) were identified as a mixture of 3′-hydroxy-4′,5′-dimethoxyflavone (**8**), 3′,4′-dimethoxyflavone (**9**) and 3′,4′,5′-trimethoxyflavone (**10**).

### 2.6. Nuclear Magnetic Resonance (NMR)

One-dimensional (^1^H-NMR, ^13^C-NMR) and two-dimensional -NMR spectra (COSY, HSQC, HMBC) spectra were recorded on Bruker Avance III 400 MHz and Bruker Avance II 200 MHz (Bruker BioSpin, Rheinstetten, Germany) spectrometers, using methanol-*d*_4_ as solvent.

### 2.7. Total Phenolic Content (TPC)

The total phenolic content of the samples was determined by the Folin–Ciocalteu method [33]. In a 96-well plate, 25 μL of extract of different concentrations (4, 2, 1 mg/mL) or standard solutions of gallic acid (2.5, 5, 10, 12.5, 20, 25, 40, 50, 80, 100 g/mL), both diluted in DMSO, were added to 125 μL of a Folin–Ciocalteu solution (10%), followed by the addition of 100 μL of 7.5% sodium carbonate. The plate was incubated for 30 min, in darkness, at room temperature. The absorbance at 765 nm was measured using a TECAN Infinite m200 PRO multimode reader (Tecan Group, Männedorf, Switzerland). All measurements were performed in triplicate, the mean values plotted on a gallic acid calibration curve, and the total phenolic content was expressed as mg equivalent to gallic acid (GAE) per gram of dry extract.

### 2.8. DPPH (2,2-DiPhenyl-1-PicrylHydrazyl) Assay

The antioxidant activity of the samples was evaluated by DPPH (1,1-diphenyl-2-picrylhydrazyl) radical scavenging activity, according to the literature [33]. For the DPPH assay, the methanol extract (concentration of stock solution 4 mg/mL) was prepared using dimethylsulfoxide (DMSO) as a solvent. 10 μL of sample were mixed with 190 μL of DPPH solution (12.4 mg/100 mL in ethanol) in a 96-well plate, then subsequently incubated, at room temperature, for 30 min in darkness. Finally, the absorbance was measured at 517 nm using an Infinite M200 Pro TECAN photometer (Tecan Group, Männedorf, Switzerland). All evaluations were performed in triplicate, while gallic acid was used as positive control (IC50 = 2.6 μg/mL). The % inhibition of the DPPH radical for each dilution was calculated using the following formula: % inhibition = [(A-B)-(C-D)]/(A-B) × 100, where A is control (without sample), B is blank (without sample, without DPPH), C is sample, and D is blank sample (without DPPH). The samples were tested at final concentrations of 200 μg/mL, 100 μg/mL and 50 μg/mL, were analyzed in triplicate, and the results were expressed as means ± standard deviation (n = 3).

## 3. Results

### 3.1. Identification of Secondary Metabolites

Based on the chromatographic analysis of the infusion of the flowers of wild *Primula veris* L. subsp. *veris* from Lake Prespa National Park, ten secondary metabolites were isolated through several chromatographic techniques (Fractionation and purification flow chart in the Appendix A) and identified through NMR spectral analysis (Appendix A): quercetin-3-O-β-glucopyranoside (**1**), isorhamnetin-3-O-β-glucopyranoside (**2**), quercetin-3-O-β-glucopyranosyl-(1 → 2)-β-glucopyranosyl-(1 → 6)-β-glucopyranoside (**3**), isorhamnetin-3-O-β-glucopyranosyl-(1 → 2)-β-glucopyranosyl-(1 → 6)-β-glucopyranoside (**4**), 3-O-neohesperidosides of kaempferol (**5**) and quercetin (**6**), kaempferol-3-O-(2″,6″-di-O-rhamnopyranosyl)-β-glucopyranoside (known as clitorin) (**7**), 3′-hydroxy-4′,5′-dimethoxyflavone (**8**), 3′,4′-dimethoxyflavone (**9**), and 3′,4′,5′-trimethoxyflavone (**10**), comparable to the literature data.

Furthermore, mass spectral analysis was performed to identify the bioactive compounds in both studied infusions of the flowers of wild and ex situ cultivated *P. veris* subsp. *veris* (LC chromatograms in the Appendix A). Fourteen compounds (Table 1) were tentatively identified by comparing the mass spectral data with the literature.

### 3.2. Determination of Total Phenolic Content (TPC) and DPPH Free Radical Inhibition

The total phenolic content was measured in the infusions of wild and cultivated *P. veris* subsp. *veris*. Results were determined using the gallic acid standard curve and expressed as equivalents thereof. The free radical scavenging activity was established by a DPPH assay. The percent reduction of the DPPH radical was calculated using the following equation: DPPH inhibition (%) = 100 − (sample/control) × 100.

The results of the determination of the TPC and DPPH free radical inhibition rates are presented in Table 2.

## 4. Discussion

The aim of this study was the phytochemical investigation and comparison of wild and ex-situ cultivated *P. veris*. subsp. *veris*.

For the first time, an infusion (prepared according to the EMA’s monograph [7]) of flowers of wild plants from the PNP, as well as an infusion of flowers of the cultivated ex situ *P. veris* subsp. *veris*, were studied.

Based on the results, it is concluded that there are significant similarities in both studied samples in terms of qualitative analysis, according to the UHPLC-HRMS study, where 12 flavonol-glycosides (mono-, di- or tri-3-O-glycosylated) and 2 methoxy flavones (chemotaxonomic markers of the genus) were detected.

A total of ten metabolites were also isolated and fully determined spectrally, of which three flavonol-glycosides [quercetin- and kaempferol- neohesperidoside (**5**,**6**) and clitorin (**7**)] are identified for the first time in the species *P. veris*.

Furthermore, the metabolites quercetin-3-O-β-glucopyranoside (**1**) and isorhamnetin-3-O-β-glucopyranoside (**2**) were identified, which have been previously found in the flower extract of *P. officinalis* (syn. *P. veris*) [13], as well as in leaves of *P. elatior* [34].

The di-glucopyranosides of quercetin and isorhamnetin, the metabolite quercetin-3-O-dirhamnosyl-hexoside, as well as the tri-glucosides of quercetin, isorhamnetin and kaempferol have been found previously in an ethanolic extract of the flowers of *P. veris* growing wild in Greece (Epirus region) [21].

The metabolite clitorin (**7**) according to the literature, has been found before in the species *P. albenesis*, *P. auricula*, *P. farinose* and *P. halleri* [20], but not in *P. veris*.

The neohesperidosides of quercetin, isorhamnetin and kaempferol have been found previously in leaves of other *Primula* species, such as *P. daonensis*, *P. hirsute* and *P. latifolia* [34,37,39,40], and now in *P. veris* for the first time.

The isolated methoxy-flavones (**8**–**10**) have been identified previously in the leaves [13,15] as well as in the flowers of *P. veris* [41].

The antioxidant capacity of the infusions of the studied samples has been determined, showing that the flowers of the wild *P. veris* expressed a fairly rich phenolic profile, while the ex-situ plant showed a lower phenolic content. Furthermore, the infusion of the flowers of the wild plant showed strong antioxidant activity (70.79% DPPH inhibition), higher than that of ex situ cultivated *P. veris* sample (44.17%), which is linked to its high phenolic load, as the related phenolic compound’s antioxidant activity is known in the literature. The higher phenolic content of wild *P. veris* is not surprising, as it is known that plants growing in natural conditions have increased requirements in order to ensure their adaptation to stress conditions (infestations by insects, animals, extreme weather conditions, etc.), which result in producing a higher percentage of secondary metabolites, such as phenolic compounds [42]. The role of secondary metabolites in the interactions with several pests and predator organisms has been the subject of several studies on *Primula* species. Theories regarding possible protecting role of various phenolic compounds for the plants have been recorded in the literature [34].

The antioxidant activity of the infusions of the studied samples has been studied in comparison with ethanolic extracts of *P. veris* flowers [1], where it was found that the TPC of cowslip in different conditions (temperature and time) was ranged between 65 and 86 mg GAE/g, showed that the wild *P. veris* from Prespa region (103.45 mg GAE/g) was richer in phenolic content, while the ex situ sample (69.66 mg GAE/g) was in the range determined previously.

## 5. Conclusions

The northwest region of Greece, where Prespa belongs, used to have an economy based on lignite mining until 2019. According to environmental policies to promote the use of natural gas and renewable energy, lignite mining has been minimized, creating a job gap in the area. Nowadays, the goal is the development of innovative agriculture activities in the region, as well as the link between agriculture and applied research, which could boost agricultural income for the local economy and create job positions [43]. In the framework of this economic analysis, the results of the current study would potentially promote the valorization of sustainable cultivation of *P. veris* in this region.

Furthermore, large amounts of plant biomass of cowslip primrose are illegally collected from wild habitats, mostly in the same geographic region, every spring during flowering period, and sold to the trade market. Therefore, different methods for the production of *P. veris* plant biomass should be adopted. The successful ex situ attempt for the production of *P. veris* subsp. *veris* by the Hellenic Agricultural Organization-Demeter, Institute of Plant Breeding and Genetic Resources in 2020, was an important achievement in the context of protecting the species from its increasing harvest, contributing to the valuable adequacy of the herbal source in our country, and ensuring significant population conservation. As is shown in this study by comparative analyses, *P. veris* subsp. *veris* of the ex situ cultivated plant material could have similar benefits, due to its similar phytochemical profile with the wild plant. Therefore, the received data is considered adequate to promote the cultivation of the plant on a larger scale in order to preserve the natural environment, as well as overall crop supplies for cowslip’s sustainable use in Greece.

With full respect to protection of plant kingdom biodiversity, which is among the most high-value challenges nowadays, the current study offers the very promising phytochemical profile of plant material of Greek native *P. veris* subsp. *veris* from the unique flora of the Lake Prespa National Park, comparing and evaluating it with flowers of the same plant after ex situ conservation and further in pilot field cultivation. The almost identical chemical profiles of the analyzed samples support the further uses of ex situ propagation and cultivation, facilitating their sustainable exploitation as food additives and therapeutic agents for future applications in the medicinal-cosmetic as well as agro-alimentary sectors.

## Figures and Tables

**Table 1 foods-12-02623-t001:** Secondary metabolites of the infusions of the flowers of wild and cultivated *P. veris* subsp. *veris* using UHPLC-HRMS in negative mode.

	RT (Min)		Compounds	*m*/*z* [-]	MS/MS Fragment Mass	Determined by
	PVPinf	PVDinf				
1.		2.56	Quercetin-3-O-β-glucopyranoside	463.08	300.03/271.02	[13,34], NMR
2.	2.63	2.65	Quercetin-3-O-β-glucopyranosyl- (1-2)-β-glucopyranosyl- (1-6)-β-glucopyranoside	787.19	300.03/271.02/255.03/243.03	[21], NMR
3.	2.63	2.63	Quercetin-3-O-dirhamnosyl-hexoside	755.20	300.03/271.02/255.03/243.03	[21]
4.	2.72	2.73	Quercetin-3-O-β-glucopyranosyl- (1-6)-β-glucopyranoside	625.14	301.03/300.03/271.02/255.03/243.03	[21]
5.	2.74	2.76	Isorhamnetin-3-O-dirhamnosyl-hexoside	769.29	315.05/300.03/271.02/243.03	[35,36]
6.	2.74	2.76	Clitorin	739.21	285.04/255.03/227.03	[35]
7.	2.80	2.86	Kaempferol-3-O-β-glucopyranosyl- (1-2)-β-glucopyranosyl- (1-6)-β-glucopyranoside	771.20	285.04/255.03/227.03	[21]
8.	2.83	2.82	Isorhamnetin-3-O-β-glucopyranosyl- (1-2)-β-glucopyranosyl- (1-6)-β-glucopyranoside	801.21	315.05/300.03/271.02/243.03	[21], NMR
9.	2.90	2.93	Quercetin-neohesperidoside	609.14	300.03/271.02/255.03/243.03	[37], NMR
10.	2.92	2.95	Isorhamnetin-3-O-β-glucopyranosyl- (1-6)-β-glucopyranoside	639.16	315.05/300.03/271.02/255.03/243.03	[21,37]
11.	3.07		Kaempferol-neohesperidoside	593.1	285.04/255.03/227.03	[20,37], NMR
12.	3.13	3.10	Isorhamnetin-neohesperidoside or Isorhamnetin-3-rutinoside	623.16	315.05/299.02/271.02/255.03/243.03	[13,21,37]
13.	5.98	5.86	3′-hydroxy-4′,5′-dimethoxyflavone	297.15	183.01	[34,38], NMR
14.	6.24	6.25	3′,4′,5′-trimethoxyflavone	311.17	183.01/174.95	[14], NMR

**Table 2 foods-12-02623-t002:** Determination of TPC and DPPH free radical inhibition of wild and cultivated *P. veris* subsp. *veris*.

Samples	TPC (mg GAE/g)	% Inhibition of DPPH^•^		
		200 μg/mL	100 μg/mL	50 μg/mL
PVP inf	103.45 ± 0.54	70.79 ± 4.11	44.00 ± 1.84	27.78 ± 1.80
PVD inf	69.66 ± 0.84	44.17 ± 1.53	24.78 ± 0.55	12.77 ± 0.43

## Data Availability

Data is contained within the article and Appendix A.

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
