# Peer review of "Comparative Phytochemical Analyses of Flowers from Primula veris subsp. veris Growing Wild and from Ex Situ Cultivation in Greece"

_foods, 2023, doi:10.3390/foods12132623_

Round 1

Reviewer 1 Report

The present study focused on the comparison of phytochemicals from flowers of wild and cultivated Primula veris and their antioxidant properties. The topic of this study fully fills in the scope of Foods. This study was designed logically and all of the results were good, but there were several suggestions for further improving the quality of the manuscript.

1.     The introduction should be more concise, and the edible values of flowers of Primula veris should be added and introduced.

2.     The contents of main flavonoids in the water extracts of flowers of wild and cultivated Primula veris could be quantified by HPLC or UHPLC.

3.     For the fractionation and purification, a flow chart is recommended.

4.     NMR data should be provided or as supplementary materials.

5.     The HRMS chromatograms of the infusions of the flowers of wild and cultivated Primula veris should be provided.

Author Response

Dear Reviewer,

thank you for your comments and suggestions.

Please find a point by point response below and weope that our answers cover adequately all comments :

  1. The introduction should be more concise, and the edible values of flowers of Primula veris should be added and introduced.

The Introduction has been revised and the values of flowers have been added in 3rd and 4th paragraph in the Introduction part.

  1. The contents of main flavonoids in the water extracts of flowers of wild and cultivated Primula veris could be quantified by HPLC or UHPLC.

The whole plan and design of this study was the qualitative analysis of the wild and the ex situ cultivated plant, on purpose to propose or not, according the results, if the ex situ cultivation could be further suggested to the area of PNP in the region of West Macedonia, which is among less rich (financially) part in the European’s Union.

  1. For the fractionation and purification, a flow chart is recommended.

A flow chart has been added in the Supplementary (Picture S8)

  1. NMR data should be provided or as supplementary materials.

All requested NMR data have been added in the Supplementary (Figure S3-S7)

  1. The HRMS chromatograms of the infusions of the flowers of wild and cultivated Primula veris should be provided.

The requested HRMS chromatograms have been added in the Supplementary (Figure S1 and S2)

Reviewer 2 Report

In the manuscript authors reported the comparative results of the phytochemical and antioxidant analysis of herbal infusions obtained from wild and ex-situ propagated cowslip flowers in Greece.

However, considering the traditional medicine usage, the authors should explain the cowslip flower's properties as food.

Other specific points:

·       Please explain in the Abstract reason for the high commercial demand.

·       The second sentence (lines 17-22) is too long and should be simplified.

·       Also, the following sentence (lines 22-29) should be more readable. Opposite to provide of all ten metabolites, consider providing only the main metabolite categories in the Abstract.

·       Are infusions obtained from Primula veris subsp veris intended to be used as herbal medicines or foods?

·       Lines 42-52 (Pages 1 and 2): Please add a reference(s) or shorten this paragraph

·       In addition to traditional medicinal use, do herbal preparations based on cowslip have nutritional properties? Are there any limitations to using cowslip as part of a regular diet?

·       Lines 118-125: This sentence is too long. Consider dividing it into 2-3 shorter.

·       Please italicize „O“ throughout the whole manuscript.

·       In addition to polyphenols, are there other bioactive compounds that contribute bioactivities and health properties of cowslip infusions?

·       Authors should discuss all factors that could influence the phytochemical compositions of samples.

·       Conclusions need to be based on the obtained results.

Minor editing of English language required. Some paragraphs/sentences should be simplified and shortened.

Author Response

Dear Reviewer,

thank you for your comments and suggestions.

Please find a point by point response below and weope that our answers cover adequately all comments :

  • However, considering the traditional medicine usage, the authors should explain the cowslip flower's properties as food.

Regarding the use of several flowers as foods and especially the edible value of the Primula species, two paragraphs have been added in the Introduction part. Several recent literature data, on the known use of cowslip flower's properties as food, have been also referred.

Other specific points:

  • Please explain in the Abstract reason for the high commercial demand.

The high commercial demand of the plant is explained in details in Lines 91-96. In the Abstract, due to word limitations, a small sentence has been added to explain this situation.

  • The second sentence (lines 17-22) is too long and should be simplified.

The sentence has been revised and has been shortened accordingly.

  • Also, the following sentence (lines 22-29) should be more readable. Opposite to provide of all ten metabolites, consider providing only the main metabolite categories in the Abstract.

The sentence has been revised and the names of the metabolites have been transferred to the Results part and they have been replaced by the chemical categories of the metabolites.

  • Are infusions obtained from Primula veris subsp veris intended to be used as herbal medicines or foods?

The Primula infusions and decoctions have very pleasant aroma and taste and were used as beverages. In most cases such beverages are also known for their traditional use for the relief of cough associated with cold. On the other hand based on the popular phrase 'Let the food be the medicine and medicine be thy food', Hippocrates (400 BC), it is obvious the importance of nutrition to prevent or cure disease. Very recently we have contributes for a chapter edited and published for Elsevier focused to such traditionally used teas and beverages and their potential to the consumers health. So, it seems that both uses beverage-food and prompter of health are linked towards the benefit of the consumers.

  • Lines 42-52 (Pages 1 and 2): Please add a reference(s) or shorten this paragraph

The paragraph has been revised and a reference has been added [Strid et al., 2020]

  • In addition to traditional medicinal use, do herbal preparations based on cowslip have nutritional properties? Are there any limitations to using cowslip as part of a regular diet?

The Primula flower's nutritional properties have been added in the Introduction part. Moreover the use of the drinks or further edible final products (herbal teas, alcoholic beverages containing extracts from the flowers etc) are safe, as till now case reports or further adverse reactions have been referred and/or published, while as it has been referred “In the USA, P. veris flowers are classified as Class 1 botanicals, which means they can be safely consumed” [McGuffin et al., 1997; EMA 2012]

Reviewer 3 Report

There is no further recommendation section. Without this section, the manuscript doesn't reflect the potential scopes of the work. Provide these sections and describe what are the study gaps which should be considered further researches. Please also provide the limitations of the work. 

Provide some figures in the manuscript i.e. in TPC and DPPH test you can provide the standard curves or results in figures. Without any figures, the manuscript looks monotonous. 

Provide the NMR spectrums and their explanations in result section where you have claimed compound isolation through NMR technique. 

These major flaws should be revised for further consideration of the manuscript. 

Can be improved

Author Response

Dear Reviewer,

thank you for your comments and suggestions.

Please find below the point by point response and we hope that our answers cover adequately all comments.

  • There is no further recommendation section. Without this section, the manuscript doesn't reflect the potential scopes of the work. Provide these sections and describe what are the study gaps which should be considered further researches. Please also provide the limitations of the work. 

Although there are numerous plant species (about 30000) considered as edible (used as food, food supplements, flavorings or colorants) worldwide, only a small portion of species has been studied until now, remaining the majority still an unexplored source. The value of Primula veris subsp. veris flowers, through chemical analysis and biological tests, has been studied and at the same time the results from the qualitative comparison of the wild with the cultivated plant, gave us the purpose to propose it for cultivation to the area of Prespa, in the region of West Macedonia, which is among less rich (financially) part in the European’s Union. A paragraph in the Introduction and another one in Conclusion have been added to cover the further recommendation section and the scope of the work, according to your suggestions.

  • Provide some figures in the manuscript i.e. in TPC and DPPH test you can provide the standard curves or results in figures. Without any figures, the manuscript looks monotonous. 

 The results from the biological test are provided in Table 2 and according to our experience on publications, there is no need to provide standard curves or figures specifically in this case where there are only two extracts for comparison.

  • Provide the NMR spectrums and their explanations in result section where you have claimed compound isolation through NMR technique. 

As requested NMR data have been added in the Supplementary (Figure S3-S7), as they are all known compounds and their explanations exist already in the literature.

Round 2

Reviewer 1 Report

no comment

Author Response

Thank you for your consideration

Reviewer 2 Report

In general, the current version is improved compared to the original manuscript. However, the manuscript requires additional changes.

The Introduction section is too long compared to the other sections. Many describing data distract the reader from the background and aims of this study. No need to detail describe ethnobotanical data. Traditional medicinal usage of Primula flowers is well known, and the EMA monograph summarizes therapeutic indications and other relevant information. Therefore, authors should focus on phytochemical compositions, factors influencing polyphenols, and potential applications of Primula flowers in nutraceuticals and food industries.

In addition to infusions, are there other data relevant to the food properties of these flowers?

Although it was previously suggested, in the revised manuscript, there is no adding content on the safety aspects of Primula flowers as foods. Since infusions were prepared following the EMA’s monograph, authors should consider contraindications,  warnings, and precautions for use as provided in this monograph. Also, should be discussed the status of this plants based on data supplied EFSA compendium of botanicals reported to contain naturally occurring substances of possible concern for human health when used in food and food supplements. These data could be supplemented in the Discussion section, which should be longer. Also, are there novelties in this study compared to others?

Conclusion data should be more concise and based on the study results. 

Throughout the whole manuscript, there are a lot of colloquial terms. The manuscript would benefit from a proofread by a native speaker to enhance the readability of the paper.

Author Response

  • The Introduction section is too long compared to the other sections. Many describing data distract the reader from the background and aims of this study. No need to detail describe ethnobotanical data. Traditional medicinal usage of Primula flowers is well known, and the EMA monograph summarizes therapeutic indications and other relevant information. Therefore, authors should focus on phytochemical compositions, factors influencing polyphenols, and potential applications of Primula flowers in nutraceuticals and food industries.

ANSWER: It was asked previously from the reviewers, to add data in the Introduction section regarding the cowslip flower's properties as food and its edible value. In order to respond adequately to the reviewers comments, 3 paragraphs were added to the Introduction section, as well as a paragraph regarding the collection area as it was a need to emphasize the unique biodiversity place in Balkan Peninsula. However, some information regarding ethnobotanical and cultural data andas well as  details from EMA’s monographs, have been deleted in this version according to reviewers’ comments.

  • In addition to infusions, are there other data relevant to the food properties of these flowers?

ANSWER: There are food companies [eg. Primula (UK)], which use Primula flowers to flavor spread cheese with a long shelf life. On the other hand there are sites (eg: https://pfaf.org/user/Plant.aspx?LatinName=Primula+veris) where it is described that: Primula flowers are used as raw (in salads), cooked or in conserves,. This species has become much less common in the past 100 years due to habitat destruction, over-collecting from the wild and farming practices. When it was more abundant, the flowers were harvested in quantity in the spring and used to make a tasty wine with sedative and nervine properties. A related species Primula elatior is listed by the Council of Europe as a natural food flavouring. Moreover, searching primula veris as edible, approx. 20 refs appears all recently published showing the interest of studying them

  • Although it was previously suggested, in the revised manuscript, there is no adding content on the safety aspects of Primula flowers as foods. Since infusions were prepared following the EMA’s monograph, authors should consider contraindications,  warnings, and precautions for use as provided in this monograph. Also, should be discussed the status of this plants based on data supplied EFSA compendium of botanicals reported to contain naturally occurring substances of possible concern for human health when used in food and food supplements. These data could be supplemented in the Discussion section, which should be longer. Also, are there novelties in this study compared to others?

Conclusion data should be more concise and based on the study results. 

ANSWER: The HMPC concluded that, on the basis of its long-standing use, primula flower preparations can be used as an expectorant (a medicine that helps bring up phlegm) for coughs associated with colds. This is the most important evidence that the plant is safe for use according to EMA monograph.

It is described well in the monograph that “Primula flower medicines should only be used in adults and adolescents over the age of 12 years. If symptoms last longer than one week or worsen during the use of the medicine, a doctor. the effectiveness of these herbal medicines is plausible and there is evidence that they have been used safely in this way for at least 30 years (including at least 15 years within the EU).”

Regarding the Contraindications it is described that: “Primula flower medicines must not be used in patients who are allergic to primula flower or to other plants of the primula species.”, which is a common sentence for the most plants.

On the same manner the Special warnings and precautions for use are:” The use in children under 12 years of age has not been established due to lack of adequate data. Caution is recommended in patients with gastritis or gastric ulcer. If dyspnoea, fever or purulent sputum occurs, a doctor or a qualified health care practitioner should be consulted.”

Moreover in the Assesement report of Primula flos it is reported in “Overall conclusions on non-clinical data” The non-clinical data on toxicology of Primula flower preparations are incomplete, but available data indicate no signals of toxicological concern, while no serious case reports have been presented until now in Eudravigilance  system

Moreover, in the list of EFSA-Q-2008-2100 it is mentioned that Primula veris “Helps to obtain a relaxation effect and regain a natural good temper. Contributes to recover physical and mental well-being”

All these infos are accessible through the monograph and EU-sites and there is no reason to include them in this manuscript as our scope is the qualitative comparison of the wild with the cultivated plant, which gave us the purpose to propose it for cultivation to the area of West Macedonia, which is among less rich ones (financially) in the European Union.

  • Throughout the whole manuscript, there are a lot of colloquial terms. The manuscript would benefit from a proofread by a native speaker to enhance the readability of the paper.

ANSWER: Thank you for this comment, following your suggestion, a native speaker has checked throughout the manuscript, and we belive that all appropriate changes have been introduced

Reviewer 3 Report

no comment

Minor corrections are encouraged i.e. there are some typos and grammatical errors.

Author Response

Thank you for this comment, following your suggestion, a native speaker has checked throughout the manuscript, and we belive that all appropriate changes have been introduced